# Stimulation of the human mitochondrial transporter ABCB10 by zinc-mesoporphrin

**Melissa Martinez, Gregory A. Fendley, Alexandra D. Saxberg, Maria E. Zoghbi** ⓘ *

Department of Molecular Cell Biology, School of Natural Sciences, University of California Merced, Merced, California, United States of America

* mzoghbi@ucmerced.edu

## Abstract

Heme biosynthesis occurs through a series of reactions that take place within the cytoplasm and mitochondria, so intermediates need to move across these cellular compartments. However, the specific membrane transport mechanisms involved in the process are not yet identified. The ATP-binding cassette protein ABCB10 is essential for normal heme production, as knocking down this transporter in mice is embryonically lethal and accompanied by severe anemia plus oxidative damage. The role of ABCB10 is unknown, but given its location in the inner mitochondrial membrane, it has been proposed as a candidate to export either an early heme precursor or heme. Alternatively, ABCB10 might transport a molecule important for protection against oxidative damage. To help discern between these possibilities, we decided to study the effect of heme analogs, precursors, and antioxidant peptides on purified human ABCB10. Since substrate binding increases the ATP hydrolysis rate of ABC transporters, we have determined the ability of these molecules to activate purified ABCB10 reconstituted in lipid nanodiscs using ATPase measurements. Under our experimental conditions, we found that the only heme analog increasing ABCB10 ATPase activity was Zinc-mesoporphyrin. This activation of almost seventy percent was specific for ABCB10, as the ATPase activity of a negative control bacterial ABC transporter was not affected. The activation was also observed in cysteine-less ABCB10, suggesting that Zinc-mesoporphyrin's effect did not require binding to typical heme regulatory motifs. Furthermore, our data indicate that ABCB10 was not directly activated by neither the early heme precursor delta-aminolevulinic acid nor glutathione, downsizing their relevance as putative substrates for this transporter. Although additional studies are needed to determine the physiological substrate of ABCB10, our findings reveal Zinc-mesoporphyrin as the first tool compound to directly modulate ABCB10 activity and raise the possibility that some actions of Zinc-mesoporphyrin in cellular and animal studies could be mediated by ABCB10.

## Introduction

The ATP-binding cassette (ABC) transporter ABCB10 was first described 20 years ago as a protein whose expression was induced by GATA-1, a transcription factor essential for normal

**Data Availability Statement:** All relevant data are within the manuscript and its Supporting information files.

**Funding:** MEZ was funded by the National Institutes of Health (1R15 GM131289-01) and University of California Merced startup funds. The

funders had no role in study design, data collection and analysis, decision to publish, or preparation of the manuscript.

**Competing interests:** The authors have declared that no competing interests exist.

erythropoiesis, and overexpression of this transporter increased hemoglobin production, suggesting that ABCB10 might mediate critical mitochondrial transport functions related to heme biosynthesis [1]. This transporter is ubiquitously expressed in human tissues, with the highest levels found in bone marrow [2]. Abcb10 knockout mice died during gestation, were severely anemic, and their erythroid cells failed to differentiate, likely due to an increase in apoptosis caused by oxidative stress [3]. Abcb10 has been shown to be essential for heme biosynthesis in both embryos and adult mice [4]. This transporter belongs to the B subfamily, which includes the widely studied multidrug resistance P-glycoprotein (ABCB1), phospholipids transporter (ABCB4), antigen peptide transporters (TAP; ABCB2 and ABCB3), and ABCB7, whose mutations have been associated with anemia sideroblastic spinocerebellar ataxia, between others [5–7]. Despite the clear essential role of ABCB10 for normal production of heme and the protection of cells against oxidative stress, the identity of the substrate(s) being transported remains unknown.

Heme biosynthesis (Fig 1A) occurs through eight sequential reactions [8] that start in the mitochondrial matrix with the production of delta-aminolevulinic acid (ALA), which is transported to the cytoplasm and converted to Coproporphyrinogen. This intermediate is then transported back to the matrix, where the final steps of heme synthesis take place. In the last reaction, the enzyme ferrochelatase (FECH) incorporates iron into the protoporphyrin IX ring (PP), forming heme. Heme must be transported to the cytoplasm where it can bind to globins and other heme-containing proteins [8, 9]. Iron is transported into the matrix through mitoferrin (Mtfn). It has been suggested that ABCB10 might be involved in the export of ALA or heme export from the matrix [10, 11]. However, no definitive evidence is available to support ABCB10 as either a heme or ALA exporter, and alternative roles have also been suggested to explain the importance of ABCB10 in heme production. ABCB10 has been found to be part of a mitochondrial heme metabolism complex, which consists of FECH, protoporphyrinogen oxidase, aminolaevulinic acid synthase-2, and other mitochondrial transporters [12]. ABCB10 has been found associated with mitoferrin-1 (Mtfn1) and FECH in mouse erythroleukemia cells, forming a complex that might synergistically integrate mitochondrial iron importation with heme biosynthesis [13, 14]. However, lentiviral gene silencing of ABCB10 in erythroleukemia cells reduced hemoglobin and decreased erythroid differentiation, suggesting that ABCB10 may exert functions early during differentiation that might not be related to mitoferrin [15]. Since the deletion of Abcb10 was observed to cause mitochondrial iron accumulation, without affecting the levels of mitochondrial ferritin, it has been suggested that ABCB10 may play a role in incorporating iron into PPIX during heme biosynthesis [4]. On another hand, knockdown of Abcb10 in cardiac myoblasts has confirmed a critical role of this protein in the heme synthesis pathway, but it was suggested that ABCB10 plays a role in early steps such as facilitating ALA production or export from the mitochondria [10]. Given the variety of putative roles proposed for ABCB10 based on experiments performed in whole animals or cells, we decided to carry out a biochemical study of the purified transporter to determine whether ALA or heme has a direct effect on the human ABCB10.

Heme is very reactive, and its concentration must be strictly regulated through gene transcription controls, heme degradation by heme oxygenase, and binding to proteins, to avoid oxidative damage to the cells [16–19]. The ferrous iron atom of free heme ($Fe^{2+}$-protoporphyrin IX, Fig 1B) easily oxidizes to ferric $Fe^{3+}$, hence a variety of heme analogs are commonly used for biochemical and physiological studies. Heme analogs can have different ions coordinated in the protoporphyrin ring, such as the oxidized iron $Fe^{3+}$ coordinated with anions (hemin, Fig 1C), the more stable metal zinc ($Zn^{2+}$-protoporphyrin IX, ZnPP, Fig 1D), or other metals. Vinyl groups ($-CH = CH_2$) in the protoporphyrin ring (Fig 1B) have chemically reactive double bonds, whereas their reduction to ethyl groups ($-CH_2-CH_3$) generates a more

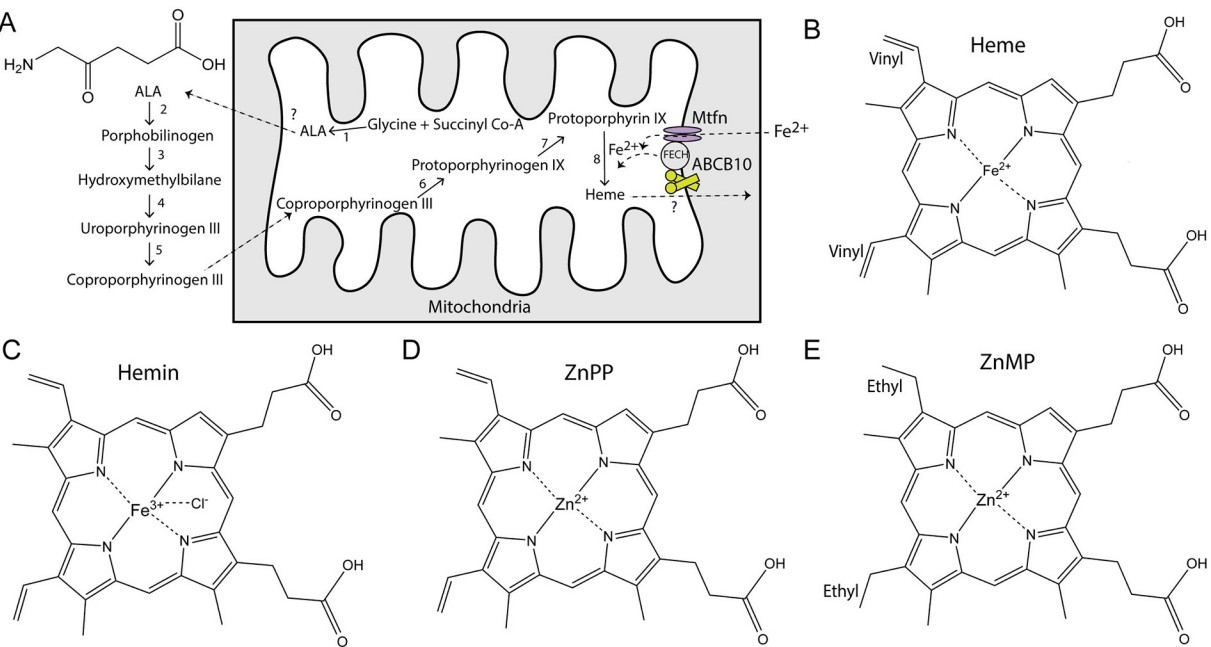

**Fig 1. Heme synthesis pathway and structures of porphyrins.** (A) Heme synthesis occurs through eight reactions that take place between the mitochondrial matrix and the cytoplasm. The first step is the formation of delta-aminolevulinic acid (ALA, structure shown in the left top corner), which is then exported to the cytoplasm through a yet unidentified mechanism. ABCB10 can form a complex with ferrochelatase (FECH) and mitoferrin (Mtfn). Once synthesized, heme is exported by an unidentified protein. (B) Structure of heme ($Fe^{2+}$ protoporphyrin). (C) Structure of hemin ($Fe^{3+}$-Cl protoporphyrin). Commercially available heme comes in the form of hemin since in free heme the iron can be quickly oxidized. (D) Structure of $Zn^{2+}$-protoporphyrin. (E) Structure of $Zn^{2+}$-mesoporphyrin; notice the presence of ethyl instead of vinyl groups in the porphyrin ring. All chemical structures were created in ChemDraw.

stable mesoporphyrin ring (Fig 1E). Zinc-mesoporphyrin (ZnMP) is a well-established heme analog [20], where the $Zn^{2+}$ is coordinated in the center of the near planar porphyrin ring (Fig 1E), as expected for $Fe^{2+}$ in heme. Heme analogs have been widely used as competitive inhibitors of heme oxygenase, can also induce heme-dependent gene transcription, and have been found to be substrates for transport proteins [21–24]. The central metal within the porphyrin ring and substituents on the ring can play a crucial role in the inhibitory potency of metalloporphyrins on heme oxygenase activity [24]. Synthetic metal-porphyrin complexes cannot be enzymatically degraded to bile pigments by heme oxygenase and *in vivo* they might have dual actions both inducing expression and inhibiting activity of this enzyme, which can lead to different biological actions depending on the nature of the metalloporphyrin [25]. In addition to their possible therapeutic roles [26–28], some metalloporphyrins can also be naturally available in certain conditions, such as occurs with the ZnPP formed by ferrochelatase when iron levels are low [29, 30]. Therefore, here we have studied the direct effect of a variety of porphyrins on ABCB10 activity.

Each subunit of homodimeric ABC transporters contains a transmembrane domain where substrates bind while they are being translocated across the phospholipid membrane and a nucleotide-binding domain that binds and hydrolyzes ATP. The exact molecular mechanisms coupling ATP hydrolysis with substrate translocation are still under investigation, but it is generally accepted that ATP binding and/or hydrolysis promote conformational changes that switch the substrate-binding pocket between inward and outward-facing conformations [31, 32]. Intriguingly, ABC transporters display ATP hydrolysis activity even in the absence of a substrate. This basal ATPase activity is commonly increased by the presence of substrate, likely

as the result of conformational changes induced by substrate binding [32]. Some ABC transporters are very promiscuous, having various pockets for binding of substrates and other molecules that can act as activators or inhibitors, with the multidrug resistance ABCB1 (P-glycoprotein) being an excellent example [33–36]. Monitoring changes in basal ATPase activity is an effective biochemical approach to help identify substrates of ABC transporters and reconstitution of the detergent purified protein into lipid nanodiscs has been successfully used to study the effect of substrate on the activity of other well known ABC transporters such as P-glycoprotein [35, 37–39]. Our data indicate that ABCB10 was not directly activated by neither the early heme precursor delta-aminolevulinic acid nor the antioxidant glutathione, suggesting that ABCB10 might not transport these molecules. ABCB10 was not activated by heme analogs like hemin or ZnPP, suggesting that heme might not be a substrate for this transporter either. We have found that the only molecule that directly increased the ATPase activity of ABCB10 was ZnMP, also considered a heme analog, indicating that ABCB10 can discriminate between molecules that share a large similarity. Future studies are needed to determine the structural bases that make ZnMP an activator of ABCB10.

## Materials and methods

### ABCB10 expression, purification, and reconstitution

Human ABCB10 without mitochondrial targeting sequence (ABCB10[152–738]) [40], with either a C-terminal (Thermo Scientific) or N-terminal poly-histidine-tag (Genescript, codon-optimized) was expressed in *E. coli* BL21-CodonPlus (DE3)-RIPL (Agilent) or Rosetta 2 (DE3) (Novagen) cells as described before [41]. Protein expression was induced with 0.5 mM isopropyl β-d-1-thiogalactopyranoside (IPTG) at 30°C, 125 rpm. Cell pellets were collected by centrifugation (4,000 g; 15 minutes in Sorvall Legend XFR, TX-1000 rotor) 3 to 4 hours after induction, resuspended in 100 mM NaCl, 10% glycerol, 20 mM Tris/HCl pH 8 plus freshly added 1 mM phenylmethylsulphonyl fluoride (PMSF), and frozen at -80°C until ready to use. Cells were lysed either with EmulsiFlex (Avestin) or by sonication after 1-hour treatment with 0.5 mg/ml lysozyme (Fisher Scientific) at 4°C with rotation. Homogenate was mixed with an equal volume of 2 M NaCl, 10% glycerol, 20 mM Tris/HCl pH 8, and 1 mM PMSF and incubated at 4°C with rotation for 10 minutes. Debris and unbroken cells were removed by low-speed centrifugation (6,000 g; 15 minutes in a Sorvall Legend XFR, F14-6x250 fixed angle rotor). The supernatant was recovered and centrifuged in a Sorvall WX ultracentrifuge (~109,000 g; 1 hour, F50L-8x39 rotor). The pellets, which contain the membrane fraction, were resuspended in 300 mM NaCl, 10% glycerol, 20 mM Tris/HCl pH 8, 20 mM imidazole, 1 mM PMSF, and 0.5 mM tris(2-carboxyethyl)phosphine (TCEP) using a Dounce homogenizer and this membrane suspension was solubilized with 1% dodecyl-maltoside (DDM, Anatrace) and 0.1% cholesteryl hemi-succinate (CHS, Anatrace) for 1 hour at room temperature with rotation. The solubilized membranes were centrifuged in a Sorvall WX ultracentrifuge (~109,000 g; 45 min, F50L-8x39 rotor) and the supernatant was incubated with Ni-NTA resin (Qiagen) for 2 hours. Column was washed with 300 mM NaCl, 10% glycerol, 20 mM Tris/HCl pH 8, 20 mM imidazole, 0.06% DDM, 0.02% CHS, 0.5 mM TCEP and protein eluted with same buffer containing 300 mM imidazole and pH 7.5. Poly-histidine-tag was cleaved with TEV protease during overnight dialysis in 100 mM NaCl, 10% glycerol, 20 mM Tris/HCl pH 7.5, 0.06% DDM, and 0.5 mM DTT, followed by a second Ni-NTA purification step. ABCB10 was reconstituted in nanodiscs [42] by mixing purified ABCB10 in detergent with *E. coli* polar or total lipids (Avanti Polar Lipids) and membrane scaffold protein MSP1D1 at a molar ratio of 1 transporter to 6 MSP1D1 and 360 lipids [41]. Detergent was removed with Bio-beads SM2 adsorbent media (BioRad). Reconstituted sample was recovered from the bio-beads and

centrifuged at 10,000 g for 10 minutes at 4˚C. An aliquot of the supernatant was injected into a 24 mL size-exclusion chromatography column (Enrich 650, BioRad) equilibrated with nano-disc buffer (100 mM KCl and 20 mM Tris/HCl pH 7.5) using an NGC Quest 10 chromatography system (BioRad). Concentration of ABCB10 in nanodiscs was estimated from Coomassie-stained gels using aliquots of known concentration of ABCB10 in detergent as reference. Empty nanodiscs, used as a control, were reconstituted similarly but adding dialysis buffer to the reconstitution mix instead of purified ABCB10. All procedures were performed at 4˚C unless indicated.

## Production of cysteine-less ABCB10

Mature human ABCB10 (without mitochondrial targeting sequence) contains 3 cysteines. Those cysteines are not conserved and they were replaced by amino acids present in ABCB10 of other species: C215S/C224L/C582G. Cysteine-less ABCB10 (ABCB10-CL; codon-optimized, Genescript) was expressed, purified, and reconstituted in nanodiscs as was described for the wild-type protein.

## Production of MsbA

The bacterial transporter MsbA was expressed, purified, and reconstituted as described before [43]. Briefly, *Salmonella typhimurium* MsbA (T561C in a C88A/C315A cysteine-less background) with an N-terminal poly-histidine-tag was expressed in BL21-CodonPlus (DE3)-RIPL (Agilent). Expression was induced with 1 mM IPTG for 4 hours at 30˚C and membrane proteins were solubilized with 1% DDM and 0.04% sodium cholate in a buffer containing 100 mM NaCl, 20 mM Tris/HCl, pH 8, 10% glycerol, 0.5 mM TCEP, and 0.5 mM PMSF. Solubilized MsbA was incubated with Ni-NTA resin, and the column was washed with 1 M NaCl, 20 mM Tris/HCl, pH 8, 10% glycerol, 0.5 mM TCEP, 20 mM imidazole, 0.06% DDM, and 0.02% sodium cholate. Protein was eluted with buffers containing 100 mM NaCl, 20 mM Tris/HCl, pH 7.5, 10% glycerol, 0.5 mM TCEP, 20 mM imidazole, 0.06% DDM, and 0.02% sodium cholate. Purified MsbA was reconstituted in nanodiscs with *E. coli* total lipids (Avanti Polar Lipids) and membrane scaffold protein MSP1D1 at a molar ratio of 1 transporter to 6 MSP1D1 and 360 lipids.

## Chemicals

All porphyrins, protoporphyrin IX (PP), mesoporphyrin IX dihydrochloride (MP), Zn (II) mesoporphyrin (ZnMP), Zn (II) protoporphyrin (ZnPP), Fe (III) mesoporphyrin IX chloride (FeMP) and hemin chloride were from Santa Cruz Biotechnology (Chem Cruz). ZnMP was also purchased from Frontier Scientific. Porphyrin stocks were made in dimethyl sulfoxide (DMSO), kept in the dark, and used within hours. The DMSO used as a solvent for the porphyrins had no effect on the activity of the transporter (S1 Fig). Reduced glutathione (GSH), oxidized glutathione (GSSG), and 5-aminolaevulinic acid (ALA) were all from Sigma-Aldrich, and their stocks were made fresh in water. Sodium hydrosulfite (Fisher Scientific), used to reduce hemin to heme, was made in water immediately before use.

## Measurements of ATPase activity

The ATP hydrolysis rate of reconstituted ABC transporters was measured by an enzyme-coupled ATPase assay that couples ATP hydrolysis with NADH oxidation monitored as a decreased absorbance at 340 nm [44]. The cocktail buffer for the assay contained 100 mM KCl, 5 mM NaCl, 20 mM Tris-HCl pH 7.5, 5 mM Na-ATP, 10 mM MgCl$_2$, 3 mM

phosphoenolpyruvate, 0.8 mM NADH, 0.5 mM TCEP plus the enzymes pyruvate kinase and lactate dehydrogenase. All reagents and enzymes were from Sigma-Aldrich. The assay was carried out on 96-well UV-transparent flat-bottom plates (Corning) and NADH absorbance was measured at 340 nm in a microplate reader (SpectroFluor, Tecan or SPECTROstar Nano, BMG Labtech) for 3 to 4 hours at 37˚C. For the assay, 0.5–1 µg of reconstituted ABCB10 were added per well. This amount of transporter allows to record a slow NADH linear decrease for an extended period of time, thus helping to confirm the stability of the protein during the assay (lost of linearly would indicate inactivation on the transporter). Given the higher ATPase rate for reconstituted MsbA [43], in control experiments only ~0.1 µg of MsbA were added per well to allow for a comparable recording time scale for both transporters. To test the effect of different drugs, 1 µL of the fresh stock was added to the well already containing 200 µl of cocktail and sample. Blanks included empty nanodiscs and nanodisc buffer with and without drug. ATPase hydrolysis rates were calculated from the slope of the linear decay, using an extinction coefficient $(\varepsilon) = 6220$ $M^{-1}.cm^{-1}$ and a pathlength (l) = 0.52 cm. The slope of the linear region of the plots was determined by a straight-line fitting using the software Origin (Microcal) or MARS data analysis software (BMG Labtech).

## Statistical analysis

ATPase rates under different conditions were analyzed by paired t-test in Origin by comparing each of the drugs to the drug-free protein. Data are presented as mean ± standard deviation of at least three independent experiments performed with independent protein purifications and reconstitutions. Individual data points of experiments are shown as empty circles.

## Results and discussion

### Effect of porphyrins on ABCB10 activity

ABCB10 is essential for heme biosynthesis [3, 4, 10, 15], and it has been suggested that this transporter plays a role either in an early or a late synthesis step (Fig 1A). Given that the basal ATPase activity of ABC transporters is increased by the presence of substrate, we decided to explore the effect of putative substrates on the activity of ABCB10. Our biological sample consists of purified ABC transporter reconstituted in lipid nanodiscs, using empty-nanodiscs and the bacterial transporter MsbA as controls (Fig 2A). We have tested the effect of metalloporphyrins that are commonly used heme analogs, and their metal-free porphyrin rings (Fig 2B). Interestingly, we found that the ATPase activity of ABCB10 had a near 70% increase in the presence of the heme analog ZnMP (179 ± 40 *vs* 305 ± 43 nmol Pi/mg protein*min; n = 7). This ZnMP-induced activation occurred in the low micromolar range ($K_m$ = 0.9 µM; Fig 2C) and was specific for ABCB10 since under the same experimental conditions the activity of the negative control bacterial ABC transporter MsbA was unaltered by the presence of ZnMP (Fig 2C, inset). The presence of $Zn^{2+}$ in the MP ring is required for this activation since metal-free MP did not produce a significant effect on the ATPase activity (Fig 2B). In control experiments, the addition of $ZnCl_2$ had no stimulatory effect on the ATPase activity of the transporter (Fig 2D). On the contrary, $ZnCl_2$ at 20 µM caused partial inhibition (Fig 2D inset). Since neither MP nor $Zn^{2+}$ could individually activate ABCB10, we can conclude that the complex ZnMP was responsible for the increased ATPase activity. On another hand, when ferric $Fe^{3+}$ ion was coordinated to the MP ring (FeMP), the activity of ABCB10 was not affected, suggesting that the presence of a coordinated divalent cation is needed for activation.

We also found that the activation of ABCB10 was exclusively observed with ZnMP since $Zn^{2+}$ coordinated with the naturally occurring PP ring (compare Fig 1D and 1E) did not affect the ATPase activity (Fig 2B). None of the tested PP compounds (metal-free PP or hemin)

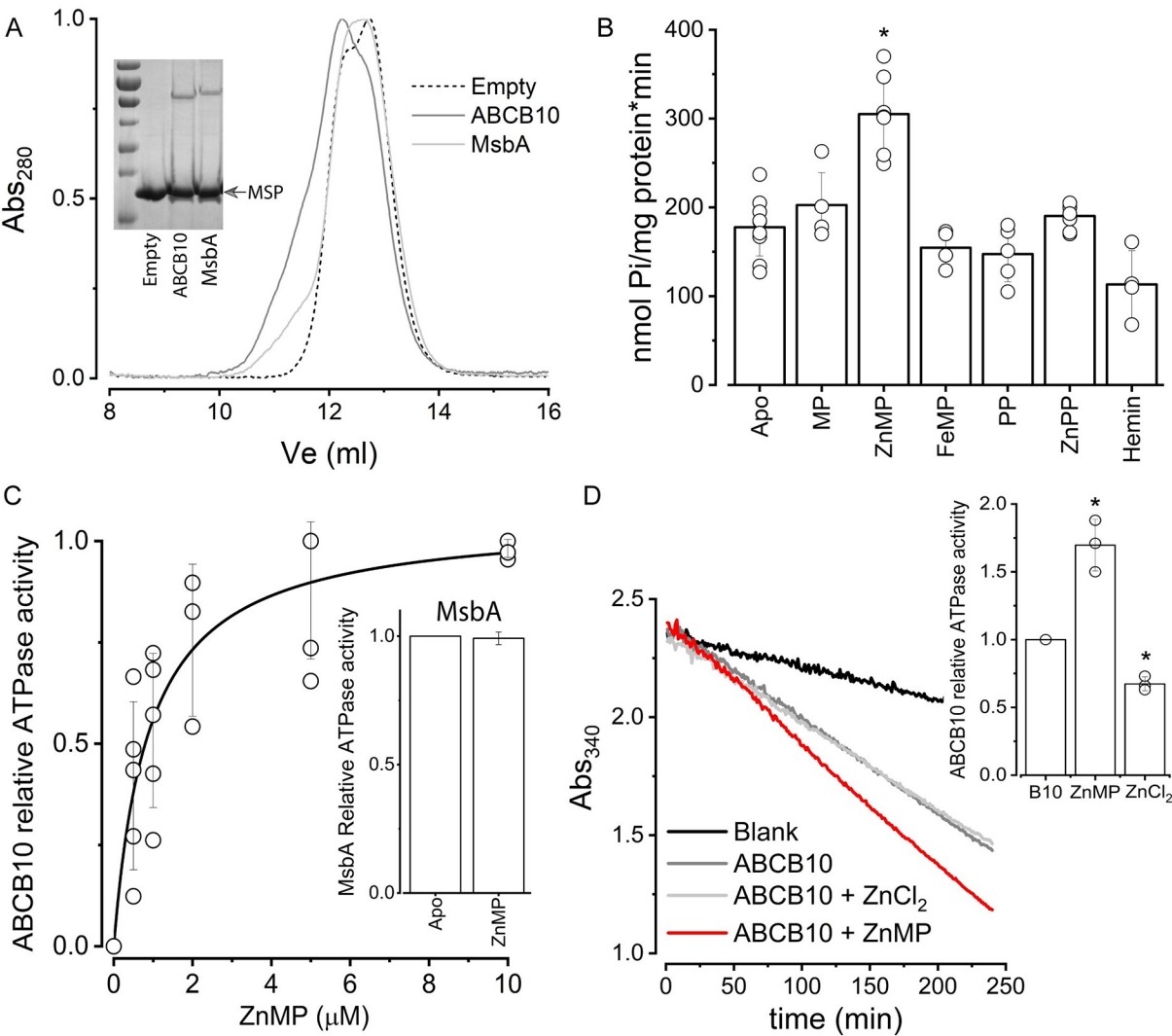

**Fig 2. Effect of porphyrins on reconstituted ABCB10.** (A) Size exclusion chromatography elution profiles of empty nanodiscs (dashed line) and nanodiscs containing either ABCB10 (dark gray) or MsbA (light gray). The shoulder around 11 ml corresponds to nanodiscs that contain ABC transporter. The inset shows a Coomassie-stained 10% polyacrylamide gel of the nanodisc preparations, with the low molecular band corresponding to MSP1D1 (MSP arrowhead) and the larger molecular weight band to the respective ABC transporter (additional information can be found in S2 Fig). (B) ATPase activity of ABCB10 in the absence (apo) or presence of 5 µM porphyrin. The increase induced by ZnMP was significant ($p = 0.05$; $n = 7$). (C) Activation of ABCB10 by increasing concentrations of ZnMP, and fitting to a Michaelis-Menten equation (black line). Inset shows the normalized ATPase activity of MsbA in the absence or presence of 5 µM ZnMP. Data represented as the mean and standard deviation of at least three independent experiments. (D) Enzyme-coupled assay of ABCB10 in presence of $ZnCl_2$. The activity of ABCB10 (dark gray curve) was increased by 5 µM ZnMP (red line, with steeper decay), but $ZnCl_2$ (20 µM, light gray line) did not induce activation of ABCB10. Inset shows the calculated ATPase rates relative to the apo protein. Data from three independent experiments, with a significant difference between the three conditions ($p = 0.05$).

induced activation of ABCB10. Lack of activation by metal-free PP was expected, since binding and/or transport of metal-free PP by ABCB10 could potentially lead to competition with ferrochelatase and negatively impact heme synthesis. However, the lack of activation by hemin or ZnPP was surprising given that they are both widely used as heme analogs, suggesting that heme might not be a substrate for ABCB10. Instead of activation, under our experimental conditions, hemin had an inhibitory effect that was very pronounced at higher concentrations (Fig 3A). However, this inhibitory effect was likely due to undesired damage

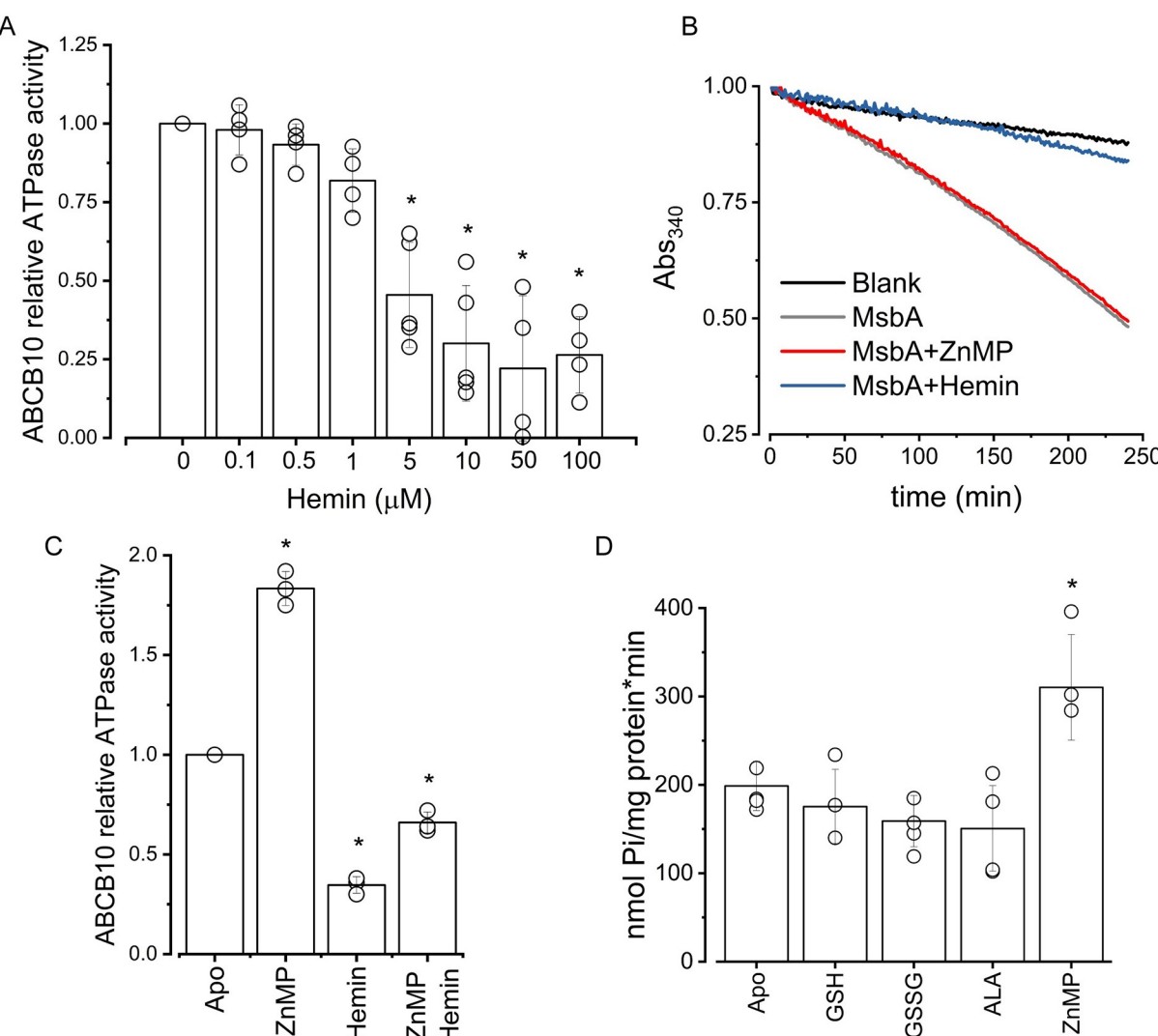

**Fig 3. Effect of hemin and other putative substrates on ATPase activity of ABCB10.** (A) Relative ATPase activity of ABCB10 at increasing concentrations of hemin. Data normalized to the apo condition. Data from at least three independent experiments, p = 0.05. (B) Representative enzyme-coupled assay of MsbA showing inhibition by 100 μM hemin (blue curve), but no effect by ZnMP (5 μM, red curve). Data normalized to the highest intensity (Absorbance value at time 0). (C) Relative ATPase activity of ABCB10 in the presence of either 5 μM ZnMP or 5 μM Hemin, and in the simultaneous presence of both porphyrins (5 μM each). Data normalized to the apo condition. Data from three independent experiments, with a significant difference between all the conditions (p = 0.05). (D) ABCB10 ATPase activity was measured in the absence (apo) or presence of either 1 mM reduced (GSH) or oxidized (GSSG) glutathione, or 0.5 mM delta-aminolevulinic acid (ALA). ZnMP was used as a positive control at a concentration of 5 μM. Data from at least three independent experiments, p = 0.05.

of lipids and proteins by hemin, which has been reported before [45], since MsbA was also deeply affected by hemin (Fig 3B). The lower activity of ABCB10 in the presence of 5 μM hemin was partially recovered by the addition of 5 μM ZnMP (Fig 3C), suggesting that the remaining active protein can still be activated by ZnMP. This result also implies that ZnMP activates ABCB10 by binding to a site that is not being occupied by hemin. Attempts to study the effect of heme by treatment of hemin with sodium hydrosulfite to promote iron reduction were unsuccessful because iron became quickly reoxidized (reduction was evident by the appearance of a vibrant red color that unfortunately returned to a dark brown during the ATPase assay). Synthetic metalloporphyrins that include heavy metal chelates of PP and

MP have been used as competitive inhibitors of heme oxygenase with various results [46]. Low micromolar concentrations (1–20 μM) of ZnPP and ZnMP inhibited hematopoiesis in animal and human bone marrow, whereas Tin ($Sn^{4+}$)-metalloporphyrins had no effect and Chromium ($Cr^{3+}$)-metalloporphyrin was lethal, suggesting that the central metal atom has an important role on the effect of metalloporphyrins [47]. Tin ($Sn^{4+}$)-metalloporphyrins are potent competitive inhibitors of heme oxygenase, but Tin ($Sn^{4+}$) mesoporphyrin (SnMP) has been shown to be 10-fold more potent than Tin ($4^{+}$) protoporphyrin (SnPP) in inhibiting heme catabolism in animal model systems, suggesting that alterations in the side chain substituents of the porphyrin ring, i.e., ethyl *vs* vinyl, can alter the ability of the synthetic heme analog to inhibit heme degradation *in vivo* [48]. Therefore, our observed differential response of ABCB10 to porphyrins seems to be shared by other proteins. Future structural and biochemical studies are needed to understand the molecular interactions that lead to ABCB10 activation only by ZnMP.

## Neither aminolevulinic acid nor glutathione activate ABCB10

Initially, ABCB10 was proposed to function as a delta-aminolevulinic acid (ALA) exporter because the reduced cellular and mitochondrial heme content caused by downregulation of this transporter in cardiac cell lines were reversed by supplementation with exogenous ALA [10]. However, silencing of Abcb10 with shRNA in murine Friend erythroleukemia cells did not affect ALA export from mitochondria, indicating that Abcb10 does not transport ALA [49]. Since an increase in basal ATPase activity induced by substrate binding is characteristic of ABC transporters [32], we decided to directly test the effect of ALA on the purified and reconstituted ABCB10. Our results indicate that a low millimolar concentration of ALA had no effect on ABCB10's ATPase activity (Fig 3D). A prior study of ABCB10 in sub-mitochondrial particles (SMPs) had shown that similar low millimolar concentrations of ALA did not affect the photolabeling of ABCB10 by 8-Azido-ATP [γ] biotin [50], suggesting that ALA is unlikely to be a substrate for the transporter. Irradiation with UV light promotes covalent binding between the azido group and ABCB10, resulting in the incorporation of the biotin label. Hydrolysis of this ATP analog would lead to the separation of the biotin label located in the γ-phosphate, decreasing the amount of biotinylated transporter. Therefore, these prior data represent an indirect measurement of hydrolysis of a labeled ATP analog by ABCB10, which might not necessarily represent the hydrolysis rate of the natural nucleotide. To our knowledge, our data represent the first direct evidence showing that the ATP hydrolysis rate by purified ABCB10 is not affected by ALA, further weakening the hypothesis of ABCB10 being an ALA exporter.

ABCB10 has also been implicated in protection against oxidative stress and it has been suggested that glutathione can be either a substrate or modulator of ABCB10 [50]. By studying ABCB10 in SMPs and its labeling by 8-Azido-ATP [γ] biotin and [α-32P] 8-azido-ATP, it was found that oxidized glutathione (GSSG) stimulated the labeled ATP hydrolysis, whereas reduced glutathione (GSH) inhibited the labeled ATP binding [50]. Therefore, we decided to test the effect of reduced and oxidized glutathione on the purified and reconstituted ABCB10. Under our experimental conditions, similar concentrations of glutathione did not affect the ATPase activity of the transporter (Fig 3D), suggesting that neither GSSG nor GSH have a direct effect on ABCB10. It is possible that the reported effect on SMPs can be the result of ABCB10 glutathionylation, as was observed after treatment with GSH-biotin [50], which is unlikely to occur in the purified protein. Our data suggest that ABCB10 is not a glutathione transporter but leave open the possibility of a regulation of the transporter by glutathionylation during oxidative stress.

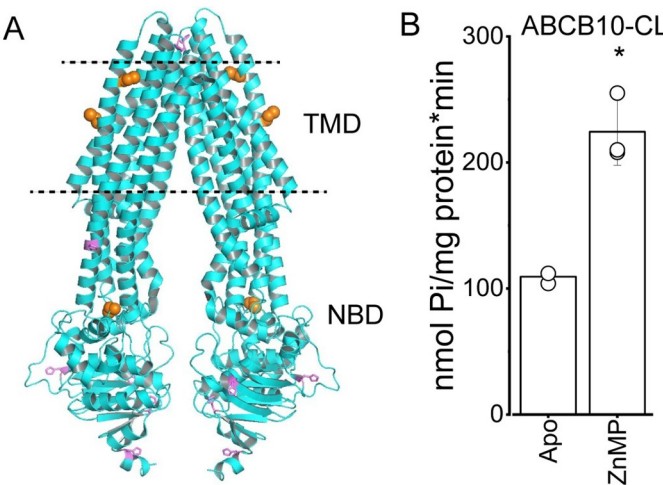

**Fig 4. Cysteine residues in ABCB10 are not necessary for ZnMP activation.** (A) Structural model of the ABCB10 homodimer (PDB 3ZDQ, both monomers in cyan), with histidine residues shown as violet sticks and cysteine residues as orange spheres. The expected phospholipid membrane borders are indicated by the dashed lines. TMD and NBD correspond to the transmembrane domain and nucleotide binding domain, respectively. Figure was created in PyMOL. (B) ATPase activity of cysteine less ABCB10 (ABCB10-CL) in the absence (apo) and presence of 5 μM ZnMP (n = 3; p = 0.05).

## Cysteines are not required for ABCB10 activation by ZnMP

Heme binding to proteins like globin and cytochromes is mediated by histidine residues but heme can also bind to cysteine residues present in heme regulatory motifs [51]. The ABC transporter ABCG2 has been shown to bind ZnMP and other porphyrins through its large extracellular loop 3, where histidine and cysteine residues are necessary for binding [21]. The mature ABCB10 protein (Fig 4A) contains three cysteines per monomer (highlighted in orange), two of them in transmembrane helix 2 and one of them in the nucleotide binding domain, far from histidine residues (highlighted in violet). To determine if ZnMP activation of ABCB10 requires the presence of cysteines we created a cysteine-less ABCB10 (ABCB10-CL) by replacing these non-conserved cysteines with amino acids (C215S/C224L/C582G) in other ABC transporters. ABCB10-CL purified and reconstituted in lipid nanodiscs was active, with an ATP hydrolysis rate near 100 nmol Pi/mg protein*min (about half of the activity for the wild-type protein). Interestingly, ABCB10-CL was efficiently activated by ZnMP (Fig 4B), which indicates that cysteines are not involved in the binding of this heme analog to the transporter. ZnMP binding likely occurs through a substrate-binding cavity in the transmembrane domains of ABCB10.

## Conclusion

Altogether, our data indicate that the heme analog ZnMP is an activator of ABCB10, suggesting that this analog could be a substrate for this orphan transporter. The ZnMP-induced activation seems to be specific, as neither other porphyrins nor the heme precursor ALA or glutathione affected the activity of the transporter. Although a transport assay is necessary to determine if ZnMP can be transported by ABCB10, this report of a molecule that can directly activate the transporter opens the door to further molecular and physiological studies important for understanding its function. In addition, ZnMP has been proposed as a therapeutic drug [26] and now we must consider the possible effects that its use can have on this

mitochondrial transporter. ZnMP appears to affect gene expression and protein levels. For example, ZnMP has been shown to decrease heme oxygenase activity, lower ALA synthase mRNA, mitochondrial ALA synthase, and plasma ALA levels in a porphyria mouse model [28]. ZnMP has also been shown to produce down-regulation of the NS5A protein (Hepatitis C Virus) by enhancing its polyubiquitination and proteasome-dependent catabolism, leading to the suggestion that ZnMP may hold promise as a novel agent to treat HCV infection [52]. ZnMP can also up-regulate heme oxygenase 1 gene expression by increasing the degradation of the Bach1 repressor in a proteasome-dependent manner [22]. Interestingly, the silencing of Abcb10 caused an increase of Bach1 on the β-globin promoter and a lower transcription of genes required for hemoglobinization, which was not reversed by hemin, suggesting that the substrate transported by ABCB10 provides an important condition or signal that optimizes hemoglobinization [49]. Therefore, we can hypothesize that transport of ZnMP (or a physiologically equivalent substrate) by ABCB10 could lead to a faster degradation of the Bach1 repressor and hence promote the transcription of hemoglobinization genes. Interestingly, human mitochondrial protoporphyrinogen oxidase (enzyme that catalyzes step 7 of heme biosynthesis, Fig 1A) can catalyze the oxidation of mesoporphyrinogen IX to mesoporphyrin IX [53]. Since mesoprophyrin IX can also be a substrate for ferrochelatase [29, 30], it would be interesting to determine whether under certain conditions ZnMP might be produced in cells.

## Supporting information

**S1 Fig. Enzyme-coupled assay of ABCB10 in presence of DMSO.** The slope of the absorbance (340 nm) decay depends on the ATP hydrolysis rate of the transporter. The blank represents a control without ABCB10 (black curves). (A) The activity of ABCB10 was not affected by addition of DMSO (2% final concentration), as it is evident from the parallel lines of ABCB10 samples with or without DMSO. This equivalent concentration of DMSO was present in the samples after porphyrins dissolved in DMSO were added to the assay mix.
(TIF)

**S2 Fig. Uncropped image of gel shown in Fig 2A.** Pre-stained protein ladder (6 μl, Thermo Scientific PI26616) or 20 μl of sample (15 μl protein + 5 μl 4X loading buffer) were loaded per well. The 10% SDS gel was run for 45 minutes at 150V and then stained with Coomassie brilliant blue. Information about the lanes is indicated in the figure. Molecular weight of the ladder bands is indicated at the left (in kDa).
(TIF)

## Acknowledgments

Sang Bao, Russell Mendiola, and Priti Kaur for assistance with general laboratory activities. Marc Liesa-Roig and Andy LiWang for critical reading of the manuscript.

## Author Contributions

**Conceptualization:** Maria E. Zoghbi.

**Formal analysis:** Melissa Martinez, Gregory A. Fendley, Alexandra D. Saxberg, Maria E. Zoghbi.

**Funding acquisition:** Maria E. Zoghbi.

**Investigation:** Melissa Martinez, Gregory A. Fendley, Alexandra D. Saxberg, Maria E. Zoghbi.

**Methodology:** Melissa Martinez, Gregory A. Fendley, Alexandra D. Saxberg, Maria E. Zoghbi.

**Project administration:** Maria E. Zoghbi.

**Supervision:** Maria E. Zoghbi.

**Validation:** Melissa Martinez, Gregory A. Fendley, Alexandra D. Saxberg, Maria E. Zoghbi.

**Visualization:** Maria E. Zoghbi.

**Writing – original draft:** Maria E. Zoghbi.

**Writing – review & editing:** Alexandra D. Saxberg, Maria E. Zoghbi.

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
