## [Decision Letter · Decision Letter 0]

14 Sep 2020

PONE-D-20-26085

Stimulation of the human mitochondrial transporter ABCB10 by zinc-mesoporphrin

PLOS ONE

Dear Dr. Zoghbi,

Thank you for submitting your manuscript to PLOS ONE. After careful consideration, we feel that it has merit but does not fully meet PLOS ONE’s publication criteria as it currently stands. Therefore, we invite you to submit a revised version of the manuscript that addresses the points raised during the review process.

As you will see from the comments of two reviewers, one is satisfied with the manuscript as it stands at present, I will therefore only comment on the reviewer who has reservations from the standpoint that, by relying solely on in vitro nanodisk assays, the relevance of these observations for the actual function of ABCB10 in mitochondria remain unclear at this stage. The reviewer thus proposes a further set of experiments in transfected cells to test the novel substrate proposed for this transporter. This would undoubtedly improve the paper's reach and I reinforce the idea. If there are limitations that do not permit these experiments to move forward, please explain in your response.

On the other side, this same reviewer asks a few clarifications on the nanodisk experiments, requesting potentially evidence of how the system operates with transporters of the same family whose function is well established for readers to compare directly the results and evaluate the new findings. I believe the criticisms offered at this level should be readily addressable by the authors.

We look forward to receiving your revised manuscript.

Kind regards,

Fanis Missirlis, Ph.D.

Academic Editor

PLOS ONE

Journal Requirements:

2.PLOS ONE now requires that authors provide the original uncropped and unadjusted images underlying all blot or gel results reported in a submission’s figures or Supporting Information files. This policy and the journal’s other requirements for blot/gel reporting and figure preparation are described in detail at https://journals.plos.org/plosone/s/figures#loc-blot-and-gel-reporting-requirements and https://journals.plos.org/plosone/s/figures#loc-preparing-figures-from-image-files. When you submit your revised manuscript, please ensure that your figures adhere fully to these guidelines and provide the original underlying images for all blot or gel data reported in your submission. See the following link for instructions on providing the original image data: https://journals.plos.org/plosone/s/figures#loc-original-images-for-blots-and-gels.

Reviewers' comments:

Reviewer's Responses to Questions

**Comments to the Author**

1. Is the manuscript technically sound, and do the data support the conclusions?

Reviewer #1: Yes

Reviewer #2: Partly

2. Has the statistical analysis been performed appropriately and rigorously? 

Reviewer #1: Yes

Reviewer #2: I Don't Know

3. Have the authors made all data underlying the findings in their manuscript fully available?

Reviewer #1: Yes

Reviewer #2: Yes

4. Is the manuscript presented in an intelligible fashion and written in standard English?

Reviewer #1: Yes

Reviewer #2: Yes

5. Review Comments to the Author

Reviewer #1: The manuscript “Stimulation of the human mitochondrial transporter ABCB10 by zinc-mesoporhin” by Martinez, Fendley, Saxberg and Zoghbi studies the effect of heme analogs, precursors and antioxidant peptides on purified human ABCB10. The authors show that the ATPase activity of ABCB10 was not activated by delta-aminolevulinic acid or glutathione. Further, the only heme-analog that was activating the ATPase activity of ABCB10 was identified to be Zn-mesoporphyrin. This identifies ZnMP as potential substrate for ABCB10, which now can be tested in future studies. All controls are provided, showing that ZnMP was not activating the bacterial transporter MsbA, ZnCL2 had no effect on the activity and also statistical analyses are provided. I have no further suggestions for the improvement of the manuscript.

I suggest acceptance of the manuscript as it stands.

Reviewer #2: Review Martinez et al PlosONE

This paper utilizes recombinant protein expression/purification and nanodisc technology to examine the activation of the Abc transporter Abcb10. They successfully express and populate Abcb10 in nanodiscs and show that the protein has ATPase activity. they then examine the effects of different porphyrins on the ATPase activity of Abcb10 and show the ZnMP increases activity by 70% whereas, other porphyrins had no affect at similar concentrations. They show that glutathiones GSH and GSSG do not affect activity and neither does the purported substrate ALA. They then mutate the cysteine residues in Abcb10 and show that it has reduced activity but that addition of ZnMP still affected activity, suggesting that these cysteine residues are important for function but are not important for ZnMP activation. These studies, although brief, inform that Abcb10 may utilize ZnMP, if present, to increase the transport activity of Abcb10. Unfortunately, this study does not go further to address this question in cell culture and instead discuss other published studies that show the ZnMP has effects on hematopoiesis making it hard to assess whether this is a physiologically important observation. To confirm that this is important, the authors could express the Wt and cysteine mutants in Abcb10 KD cells in the presence or absence of ZnMP and determine if it has any effects on hemoglobinization. If it does, then this would provide stronger support for ZnMP as a relevant activator of Abcb10.

Other critics

1. The flow of the paper is unusual with the authors referring to fig 3A in the introduction. If the point of 3A is to introduce the structures of Abcb10, perhaps it should be either included in figure 1 or referenced in the results section. Either way, both figure 1 and figure 3A are already published information and do not advance the field, thus, they do not seem unnecessary for the reader.

2. It would be nice to see if a Cysteine-less Abcb10 can complement the loss of Abcb10 in cell culture.

3. The authors use bacterial Abc transporter MsbA as a negative control. Is there a positive control Abc transporter that could be used to show that it can be activated by a known substrate but that the substrate does not affect Abcb10? Possible Positive control would be Abcb6, which is proposed to be a porphyrin transporter

4. What is the kinetics of MsbA activity compared to Abcb10? Is it a very effective ATPase?

5. It would be interesting to know if heme addition blocks the enhanced activity by ZnMP?

Minor critics

Line 171 “used to reduced hemin to heme” should be “used to reduce hemin to heme”

Line 242 “mesoporphyrin have been shown to be” should be “H\\has been shown to be”

Sup fig 1 shows a time course of absorbance change. Seems rather slow…..on the order of several hours to get 50% reduction. Is this a normal rate for an Abc transporter? Some comment is needed

Sup fig 2 atpase activity inhibited by increasing hemin - # of replicates? The graph looks like there may be only 2 replicates at some time points. If this is the case, the scientific rigor seems to be lacking for this supplemental figure.

Supplemental figures seem import – not sure why they are supplemental and the reader would recommend moving into the results section as figures.

Previous studies with Glutathione were done in submitochondrial particles which would be more reflective of what happens in vivo and should be mentioned as a caveat to the nanodisc experiments, which do not have potential activation of partner proteins.

6. PLOS authors have the option to publish the peer review history of their article (what does this mean?). If published, this will include your full peer review and any attached files.

Reviewer #1: No

Reviewer #2: No

---

## [Author Response · Author response to Decision Letter 0]

2 Nov 2020

Response to Reviewers

Reviewer #2: 

This paper utilizes recombinant protein expression/purification and nanodisc technology to examine the activation of the Abc transporter Abcb10. They successfully express and populate Abcb10 in nanodiscs and show that the protein has ATPase activity. they then examine the effects of different porphyrins on the ATPase activity of Abcb10 and show the ZnMP increases activity by 70% whereas, other porphyrins had no affect at similar concentrations. They show that glutathiones GSH and GSSG do not affect activity and neither does the purported substrate ALA. They then mutate the cysteine residues in Abcb10 and show that it has reduced activity but that addition of ZnMP still affected activity, suggesting that these cysteine residues are important for function but are not important for ZnMP activation. These studies, although brief, inform that Abcb10 may utilize ZnMP, if present, to increase the transport activity of Abcb10. Unfortunately, this study does not go further to address this question in cell culture and instead discuss other published studies that show the ZnMP has effects on hematopoiesis making it hard to assess whether this is a physiologically important observation. To confirm that this is important, the authors could express the Wt and cysteine mutants in Abcb10 KD cells in the presence or absence of ZnMP and determine if it has any effects on hemoglobinization. If it does, then this would provide stronger support for ZnMP as a relevant activator of Abcb10.

Response: we would love to determine the effect of ZnMP in cells. However, performing such experiments in cultured mammalian cells is beyond the current capabilities in our laboratory since we do not have the necessary equipment nor expertise. We are a small new laboratory (Maria E. Zoghbi is an Assistant Professor) in a relatively new campus (UC Merced was established only 15 years ago) that still lacks many resources commonly available in more established research institutions. UC Merced is in the California Central Valley, with no nearby facilities from other institutions. In addition, we do not have a collaborator currently available to help us with such experiments. Our goal with this article is to report ZnMP as a new tool for the study of ABCB10 and we hope this information will open the doors for future research, including studies at the cellular level. We do intend to expand our research capabilities once we can secure the space and financial resources needed to establish our mammalian cell culture room. 

Since we are only using the transporter reconstituted in nanodiscs as our experimental system to study the effect of drugs on the protein, we have now included in the introduction (page 5) some references to studies performed using a similar approach for the well-known multidrug ABC transporter P-glycoprotein: “Monitoring changes in basal ATPase activity is an effective biochemical approach to help identify substrates of ABC transporters and reconstitution of the detergent purified protein into lipid nanodiscs has been successfully used to study the effect of substrate on the activity of other well-known ABC transporters such as P-glycoprotein [35, 37-39].

Other critics

1. The flow of the paper is unusual with the authors referring to fig 3A in the introduction. If the point of 3A is to introduce the structures of Abcb10, perhaps it should be either included in figure 1 or referenced in the results section. Either way, both figure 1 and figure 3A are already published information and do not advance the field, thus, they do not seem unnecessary for the reader.

Response: We completely agree with the observation of unusual flow and we have therefore removed the reference to fig 3A from the introduction. However, we do think that having a figure showing the structure of many of the molecules we are testing (Fig 1) to the reader facilitates the understanding of the article. In a similar way, Fig 3A (Fig 4A in the new version) is based on an available crystal structure, but it helps the reader to visually locate the cysteine residues that are being mutated in the transporter. 

2. It would be nice to see if a Cysteine-less Abcb10 can complement the loss of Abcb10 in cell culture.

Response: We absolutely agree. Unfortunately, as mentioned above, we do not have the capability to perform experiments in mammalian cell cultures.

 3. The authors use bacterial Abc transporter MsbA as a negative control. Is there a positive control Abc transporter that could be used to show that it can be activated by a known substrate but that the substrate does not affect Abcb10? Possible Positive control would be Abcb6, which is proposed to be a porphyrin transporter.

Response: We used MsbA as a negative control since we have priorly studied this transporter and we can express and purify this protein in our laboratory using E. coli. Likewise, as we stated in the methods, we have expressed and purified ABCB10 using E. coli by a procedure we have developed, and since that article has been recently accepted (Saxberg et al, https://doi.org/10.1016/j.pep.2020.105778), we are now citing that work in the revised version of this manuscript (reference #41, in page 6-7 of the method section). In that article we show that ABCB10 produced in E. coli is properly folded and functional, similarly to the protein produced in insect cells. However, the available protocols for expression and purification of ABCB6 use mammalian cells and/or Pichia pastoris. Once again, unfortunately, we do not currently have the capability of using those expression systems in our laboratory. Therefore, purified ABCB6 was not available to be included as a positive control for some of the porphyrins in our experiments. Nevertheless, we have taken precautions to use freshly prepared stocks of all porphyrins, all of them were purchased from the same manufacturer, and the stocks were tested simultaneously during the same experiment. Under those conditions, ABCB10 was activated by ZnMP but not by the other porphyrins. 

4. What is the kinetics of MsbA activity compared to Abcb10? Is it a very effective ATPase?

Response: We appreciate this very relevant question from the reviewer. The ATPase activity of MsbA is about ten times higher than that of ABCB10 and we have now added information in the methods to address this issue (page 8-9). The low ATPase activity of ABCB10 is typical for human ABC transporters and has also been reported by Shintre et al. (DOI: 10.1073/pnas.1217042110). This comparison of ATPase rates is also directly related to the reviewer’s comment about Sup fig 1. The rate of the Abs340 decay could be faster if we add more protein per well. However, we intentionally design the experiments to record the activity for a long period of time as a way to confirm the protein’s stability during the assay. We want to be sure we are measuring the response of a stable protein, capable of withstanding several hours of activity at 37oC. Since the ATPase activity of MsbA is higher, we use less MsbA for the assay, so the time scales can be comparable. If we use too much transporter, the Abs340 will decrease to zero in a few minutes and we will miss important information. 

5. It would be interesting to know if heme addition blocks the enhanced activity by ZnMP?

Response: this is certainly an important question and we made additional experiments following the great reviewer’s suggestion. The new data is available in Fig 3C of the revised version. Page 11 now includes the following text: “The lower activity of ABCB10 in the presence of 5 �M hemin was partially recovered by addition of 5 �M ZnMP (Fig 3C), suggesting that the remaining active protein can still be activated by ZnMP. This result also implies that ZnMP activates ABCB10 by binding to a site that is not being occupied by hemin.” 

Minor critics

Line 171 “used to reduced hemin to heme” should be “used to reduce hemin to heme”

Line 242 “mesoporphyrin have been shown to be” should be “H\\has been shown to be”

Response: We apologize for the errors and deeply appreciate the careful revision made by the reviewer. Both mistakes have now been corrected in the revised version.

Sup fig 1 shows a time course of absorbance change. Seems rather slow…..on the order of several hours to get 50% reduction. Is this a normal rate for an Abc transporter? Some comment is needed

Response: Thank you for the comment. Please refer to point #4 above.

Sup fig 2 atpase activity inhibited by increasing hemin - # of replicates? The graph looks like there may be only 2 replicates at some time points. If this is the case, the scientific rigor seems to be lacking for this supplemental figure.

Supplemental figures seem import – not sure why they are supplemental and the reader would recommend moving into the results section as figures.

Response: We have followed the reviewer’s recommendations about the supplemental figures. We moved former S1 Fig (part B) to the main text (found now as Fig 2D, and new complementary data is presented in the inset). Former S2 Fig is now part of Fig 3 of the revised version, and it now includes new data of experiments done at different hemin concentrations. It was difficult for us to find a good place for S1 Fig in the main text without disrupting the current organization, so we decided to leave that plot as supplementary information. Text has been modified accordingly to take into consideration the new distribution of figures (all modifications can be followed as red track changes). We have also included the following statement in the methods (page 9): “Data are presented as mean ± standard deviation of at least three independent experiments performed with independent protein purifications and reconstitutions.”

Previous studies with Glutathione were done in submitochondrial particles which would be more reflective of what happens in vivo and should be mentioned as a caveat to the nanodisc experiments, which do not have potential activation of partner proteins.

Response: we would like to mention that in the original version we did acknowledge the prior experiments done in submitochondrial particles and the fact that we cannot have glutathionylation of the purified protein. The statement, now in page 13-14 of the revised version, says: “Under our experimental conditions, similar concentrations of glutathione did not affect the ATPase activity of the transporter (Fig 2D 3D), suggesting that neither GSSG nor GSH have a direct effect on ABCB10. It is possible that the reported effect on SMPs can be the result of ABCB10 glutathionylation, as was observed after treatment with GSH-biotin [50], which is unlikely to occur in the purified protein. Our data suggest that ABCB10 is not a glutathione transporter but leave open the possibility of a regulation of the transporter by glutathionylation during oxidative stress.

---

## [Editor Report · Decision Letter 1]

9 Nov 2020

Stimulation of the human mitochondrial transporter ABCB10 by zinc-mesoporphrin

PONE-D-20-26085R1

Dear Dr. Zoghbi,

I would like to congratulate you for this paper and wish you well in your efforts to build cell culture facilities for extending your experiments in this area.

We’re pleased to inform you that your manuscript has been judged scientifically suitable for publication and will be formally accepted for publication once it meets all outstanding technical requirements.

Kind regards,

Fanis Missirlis, Ph.D.

Academic Editor

PLOS ONE
---

## [Editor Report · Acceptance letter]

16 Nov 2020

PONE-D-20-26085R1 

Stimulation of the human mitochondrial transporter ABCB10 by zinc-mesoporphrin 

Dear Dr. Zoghbi:

I'm pleased to inform you that your manuscript has been deemed suitable for publication in PLOS ONE. Congratulations! Your manuscript is now with our production department. 

Kind regards, 

on behalf of

Dr. Fanis Missirlis 

Academic Editor

PLOS ONE